# Comparison of Fruit Texture and Storage Quality of Four Apple Varieties

**DOI:** 10.3390/foods13101563

**Published:** 2024-05-17

**Authors:** Xiaoyi Ding, Yajin Zheng, Rongjian Jia, Xiangyu Li, Bin Wang, Zhengyang Zhao

**Affiliations:** State Key Laboratory of Crop Stress Biology for Arid Areas, College of Horticulture, Northwest A&F University, Xianyang 712100, China; dxy091399@163.com (X.D.); 18748368366@163.com (Y.Z.); jrj16882021@163.com (R.J.); 18729582336@163.com (X.L.); wb765198@163.com (B.W.)

**Keywords:** cell wall, softening and ripening, storage quality, correlation analysis

## Abstract

Fruit texture and storage properties of various apple varieties exhibit significant variation. The rate of fruit softening post-harvest plays a crucial role in determining fruit quality and shelf life. This research utilized four apple varieties as test subjects to investigate the internal factors influencing fruit texture changes among different varieties. By monitoring changes in relevant physiological indicators during the post-harvest texture softening process, the study examined fruit quality, cell wall material content, hydrolase activity, and gene transcription levels during storage of ‘Orin’, ‘RX’, ‘RXH’, and ‘Envy’ apples. Initial fruit softening was primarily linked to heightened post-harvest fruit respiration intensity, ethylene production, and rapid amylase activity. Subsequent softening was associated with increased activity of water-soluble pectin (WSP), cellulose (CEL), and other hydrolases. With the extension of the storage period, the fruit cells of the four varieties became more loosely arranged, resulting in larger intercellular gaps. Variations in WSP and cellulose content, CEL activity, and relative expression of *Mdβ-gal* were observed among the different apple varieties, potentially accounting for the disparities in fruit texture.

## 1. Introduction

In China’s fruit market, the apple industry holds significant importance, contributing to over half of the world’s total production volume and planting scale [1]. With consumers increasingly prioritizing health, apples have garnered special interest for their high levels of health-promoting compounds like phenols [2], flavonoids [3], and other substances. Fruit texture is among the four key quality attributes of fruit, alongside appearance, flavor, and nutritional properties [4]. Fruit texture, as an important part of fruit quality, directly determines the taste and storage of fruit [5]. Texture serves as a valuable indicator of the inherent quality of apple fruit, often assessed through firmness, crispness, and juiciness [6]. Fruit firmness is a crucial intrinsic quality that influences consumer preferences, processing techniques, and shelf life. Apples with a soft texture are more prone to mechanical damage and pathogenic infections during transportation and storage [7]. Firmness is a crucial factor in determining the texture of fresh fruit [8], and it plays a vital role in influencing consumer satisfaction, post-harvest storability, shelf life, as well as packaging and transportation [9]. Fruit ripening is characterized by significant changes in firmness, which are directly related to texture. This crucial aspect of fruit structure is influenced by the modification, disassembly, and reduction of intercellular adhesion of primary cell wall (CW) polysaccharides. The plant cell wall is primarily made up of a complex network of polysaccharides such as pectin, hemicellulose, cellulose, and lignin, along with structural proteins. Understanding the softening mechanism during fruit development and ripening is essential for maintaining the economic quality of fruits, and this area of research is gaining increasing attention [10].

Softening is indicative of ripeness in the majority of fruits. The ripening and softening of fruit is a tightly regulated and irreversible programmed process. This process involves heightened respiration, elevated ethylene production, degradation of cell wall polysaccharides, and breakdown of starch as the main physiological and biochemical changes [11]. Yang et al. conducted a study comparing the pectin content in apple cultivars with varying textures. The researchers observed that crispy-fleshed cultivars exhibited a higher concentration of water-soluble pectin, whereas less firm fruits showed a higher content of ion-bound pectin [12]. In loquat, it has been discovered that metabolic changes, such as those involving pectin and cellulose, play a significant role in fruit texture during development. The pectin component in peach pulp decreases rapidly during the ripening process, while the water-soluble pectin content increases rapidly, leading to a quick softening of the fruit [13]. In the development and maturity of four different apple cultivars, changes in cell wall hydrolase activity and related gene expression levels have a significant impact on fruit firmness [14].

During the process of fruit softening, the modification of polysaccharides necessitates the involvement of multiple enzymes. One such enzyme, polygalacturonase (PG), is crucial for the breakdown of pectin [15]. Research has shown that as tomato fruits ripen, the activity of polygalacturonase increases gradually, leading to a decrease in fruit firmness over time [16]. Pectin methylesterase (PME), also known as pectinesterase (PE), plays a role in remodeling and breaking down pectin. During the post-harvest storage period of loquat, it was observed that the activity of PME and its gene expression levels increased as the storage period extended. A comparison of pear cultivars with varying storage properties revealed that the PME activity of both cultivars decreased over time, although the rate of decline differed between the cultivars [17]. Wolf discovered a strong correlation between PME activity and alterations in cell wall composition throughout the process of fruit ripening [18]. Galactosidase plays a crucial role in determining the tightness and firmness of the cell wall in fruits, influencing cell recognition and ultimately affecting the texture of the pulp [19]. Previous studies have demonstrated the presence of β-Gal activity in various climacteric fruits like pears [20] and peaches [21], suggesting a potential ethylene-regulated isomer that contributes to the softening process. Research on Fuji and Golden Delicious fruits revealed significant differences in β-Gal activity during storage [19]. Additionally, enzymes such as phenylalanine ammonia lyase (PAL), cinnamyl alcohol dehydrogenase (CAD), and peroxidase (POD) are essential in the lignin biosynthetic pathway.

‘Orin’ is a classic apple variety from Japan. It is characterized by its yellow-green color, smooth thin skin, and sweet flesh. ‘Envy’ is a new late-maturing apple variety originating from New Zealand. It is known for its firm texture, crispness, juiciness, and a balanced sweet and tart flavor profile, making it particularly appealing to Asian palates. ‘RX’ and ‘RXH’ are late-maturing new varieties selected by Northwest A&F University on the basis of the parents, ‘Cripps Pink’ and ‘Fuji’. ‘RX’ share the yellow-green outer peel color with ‘Orin’, and our initial research has identified a distinct fragrance in ‘RX’ [22]. The fruit of ‘RXH’ is medium in size, with smooth skin, fine and crisp flesh, and high quality. ‘RX’ and ‘RXH’ exhibit superior appearance quality, storage stability, and flavor characteristics compared to their parent varieties. In order to understand the texture changes during storage of different apple cultivars, it is crucial to investigate the compositional and structural alterations of the cell wall (CW) and its interaction with various enzymes that contribute to its degradation. While previous research has primarily examined texture changes during fruit development, there is a notable lack of knowledge regarding texture changes during post-harvest storage [11]. Therefore, this study aimed to assess fruit quality indicators, CW components, and contents of fruits from four apple cultivars during storage. Additionally, changes in gene expression and enzyme activities associated with cell wall metabolism during storage were also analyzed. The findings from this study will shed light on the physiological processes and molecular mechanisms underlying fruit texture changes in diverse apple cultivars. By comparing these different varieties, we aim to identify the main factors influencing the variation in softening speed among apple varieties. This research will serve as a basis for investigating the internal regulatory factors and molecular mechanisms underlying fruit texture changes in different apple varieties, ultimately offering insights for the selection and breeding of superior storage varieties.

## 2. Materials and Methods

### 2.1. Plant Materials

Four different apple cultivars, ‘Orin’, ‘Ruixue’, ‘Envy’, and ‘Ruixianghong’, were collected from the experimental station (35°02′ N, 109°06′ E; 908 m altitude) of Northwest A&F University in Baishui County, Shaanxi Province. These apples were of the same age and growth period, grafted on M26 rootstocks, and planted in a 1.5 m × 4 m plot. They were harvested at their commercial maturity stages and selected based on consistent size, maturity, shape, and color, free from damage or pests. After harvest, the fruits were transported to Northwest A&F University on the same day. Following a 24 h pre-cooling period, the apples were stored at (4 ± 1) °C and (85 ± 5)% relative humidity for four months. Various measurements were taken every two weeks, with fruit samples being cut and frozen for further analysis. Each experiment involved eight randomly selected fruits, with each treatment repeated three times.

### 2.2. Measurement of Physiological Characteristics

Determination of fruit weight loss: Over four months, the weight loss of two groups of fruits was measured every two weeks and expressed as a percentage. The weight loss was calculated using the following formula:Weight loss (%) = (initial weight − final weight)/initial weight × 100

The firmness of the fruit was assessed by employing a TMS-Pilot precision texture analyser (TL-Pro Test System, FTC, Atlanta, GA, USA) with a flat probe measuring 10 mm in diameter at the equatorial region of each fruit in opposing directions. Each measurement was conducted at a velocity of 1.0 mm s^−1^, encompassing pre-test and post-test parameters, with two technical replicates performed to a depth of 8 mm within 5.0 s. The determination of soluble solids content (SSC) and titratable acidity (TA) was conducted using juice extracted from randomly chosen fruits of each cultivar. SSC was gauged utilizing a brix refractometer (Atago PR-101 R, Tokyo, Japan), while TA was quantified by titrating with NaOH (0.1 mol/L) and expressed as a percentage (%) of malic acid. Each group comprised three replicates for statistical robustness.

Ethylene release rate was analyzed using gas chromatography [23]. Respiration rate was measured using a respiration rate meter [24]. The release of ethylene and respiration rate were measured in terms of the amount of CO_2_ and ethylene generated per kilogram of fresh weight in a second, respectively. Each experimental procedure was done three times, and each replication consisted of 10 pieces of fruit.

The starch content and amylase activity in the fruit were measured using spectrophotometric methods. The levels of ascorbic acid, catalase, peroxidase, and malondialdehyde in apple fruits were determined following the protocol outlined by He [25].

### 2.3. Analysis of CWMs

The procedure to extract cell wall polysaccharides followed Melton [26]. Sequential extraction was conducted to obtain water-soluble pectin (WSP), chelator-soluble pectin (CSP), ion-soluble pectin (ISP), and hemicellulose. Pectin concentration was assessed using the carbazole-ethanol technique, and cellulose concentration was determined using the weight-based method. 

### 2.4. Determination of Cell Wall Degradation-Related Enzyme Activities

PG and CEL activities were determined by the DNS colorimetric method, which was slightly modified from our previous study [11]. The measurement of β-gal activity was performed through the hydrolysis of nitrogalactoside and was quantified at a wavelength of 540 nm. The determination of PL activity was carried out following the protocol outlined by Payasi, with quantification performed at 550 nm [27]. The measurement of β-gal activity was conducted using the method described by Payasi, and quantification was performed at 550 nm [27].

### 2.5. Morphology Analysis of Apple Fruit Cell Wall

The morphology of the apple pulp cell wall was analyzed by transmission electron microscope (TEM) according to the method described by Li [11]. Small cuboids at the equator of the apple flesh (5 × 2 × 1 mm) were cut, fixed with 4% glutaraldehyde at 4 °C overnight, washed three times by PBS, and fixed in 1% osmium tetroxide for 2 h. Then, they were washed five further times with PBS. After their dehydration using a series of graded ethanol, the samples were saturated in LR White resin at 55 °C for more than two days. Finally, ultrathin sections (70 nm in thickness) were cut using an ultramicrotome (Ultracut-R, Leica, Wetzlar, Germany) and were viewed under a TEM (JEM 1230, JEOL, Tokyo, Japan).

### 2.6. Determination of Relative Gene Expression

The apple pulp samples were processed for total RNA extraction using the Quick RNA isolation kit (Tiangen, Beijing, China). The All-in-One First-Strand cDNA Synthesis SuperMix with gDNA Eraser (TransGen Biotech, Beijing, China) was utilized for cDNA reverse transcription. A RT-qPCR was conducted using the ABI7500 System and SYBR Green Master Mix (Vazyme, Nanjing, China). Actin was employed as an internal control in the experiment. The RT-qPCR primers can be found in Appendix A.

### 2.7. Statistical Analysis

To identify statistically significant differences between samples, we employed the Student’s *t*-test (*p* < 0.05 indicated a significant difference). To assess statistically significant differences among multiple samples, we utilized Tukey’s one-way analysis of variance (ANOVA) via SPSS (IBM Corporation, New York, NY, USA). Graphs were generated using GraphPad Prism 9.0 (GraphPad Software, San Diego, CA, USA). All reported values represent the mean ± standard error (SE) derived from three biological replicates.

## 3. Results

### 3.1. Physiological Characteristics of Four Apple Varieties during Storage

Fruit firmness plays a crucial role in assessing the storability of fruits and evaluating the impact of storage conditions on fruit quality. As shown in Figure 1A, the firmness of ‘Orin’, ‘RX’, ‘Envy’, and ‘RXH’ on the day of harvest were 7.63, 8.62, 8.79, and 8.82 kg cm^2^, respectively. Although there was a general decrease in fruit firmness over the storage period, ‘RX’ consistently maintained significantly higher fruit firmness compared to ‘Orin’. The rate of decline in firmness also differed significantly between the two cultivars. After 56 days of storage, ‘RXH’ and ‘Envy’ exhibited significantly higher fruit firmness levels than ‘Orin’ and ‘RX’, with a similar trend observed thereafter. By the end of the 112-day storage period, the fruit firmness of ‘Orin’ had decreased by 31.6% from harvest, while that of ‘RX’ had only decreased by 15.8%. ‘RXH’, on the other hand, experienced a minimal decrease of 6.0% in fruit firmness. Furthermore, the weight loss rate is a crucial indicator affecting fruit appearance quality and storage stability. The weight loss rates of the four cultivars gradually increased over time, with ‘RX’ showing a slower increase compared to ‘Orin’ after 56 days of storage. By the end of the 112-day storage period, ‘RX’ had a weight loss rate of only 5.08%, while ‘Orin’ fruits had a much higher rate of 10.62%. Overall, ‘Orin’ exhibited the highest weight loss rate during storage, followed by ‘RX’, ‘Envy’, and ‘RXH’, in that order. ‘RXH’ had the lowest weight loss rate at 3.70% (Figure 1B).

SSC and TA are crucial metrics for assessing fruit quality and are closely linked to the fruit’s storage characteristics. Over time in storage, the changes in SSC for the four cultivars followed relatively similar patterns. Initially, there was a gradual increase post-harvest, followed by a slow decrease (Figure 1C). ‘Orin’ recorded its highest SSC value of 16.71% after 28 days of storage, maintaining stability between 15.76 and 16.12% in later stages. In comparison, ‘RX’ reached its peak SSC value of 18.19% after 56 days of storage, with a content of 17.06% after 112 days. Notably, ‘RXH’ exhibited significantly higher SSC levels than ‘Envy’ in the early storage period (0–42 d), though the difference became insignificant after 56 days. By the end of 112 days, ‘RXH’ had a notably higher SSC content than ‘Envy’. As depicted in Figure 1D, the decline in TA for ‘Orin’ was notably more pronounced compared to the other cultivars, with the lowest decrease at 0.137%. ‘RX’ experienced only a 32% reduction in TA during storage, while ‘Envy’ and ‘RXH’ saw decreases of 34.2% and 29.6%, respectively.

The respiration rate and ethylene release of the four fruit cultivars during storage exhibited a typical climacteric pattern, characterized by an initial increase followed by a decrease (Figure 2). The peaks in respiration and ethylene release occurred on the 28th day of storage. Notably, ‘Orin’ displayed significantly higher peaks in both respiration rate and ethylene release compared to the other cultivars. Specifically, the peak ethylene release values for ‘Orin’ and ‘RX’ were 58.42 and 41.58 μL kg^−1^ S^−1^, respectively, with the former being 28.8% higher than the latter (Figure 2A). Additionally, the respiration rate and ethylene release rate of ‘RXH’ were notably lower than those of ‘Envy’ (Figure 2B).

At harvest, the starch content of ‘Orin’ was significantly greater than that of ‘RX’, and the starch content of ‘Envy’ was significantly higher than that of ‘RXH’ (Figure 3A). The starch content of all groups decreased significantly by the 14th day after harvest. Furthermore, the AM activity of the fruits exhibited a pattern of initial increase followed by a decrease during storage, peaking on the 14th day after harvest. The AM peak value of ‘Orin’ was notably higher than that of ‘RX’, with the former being 25.68% higher than the latter. Similarly, the AM peak of ‘Envy’ surpassed that of ‘RXH’, with values of 7.045 and 6.46 U g^−1^, respectively (Figure 3B).

### 3.2. Measurement of Antioxidant Activity of Four Cultivars of Fruit during Storage

As fruit naturally ages, the hydrogen peroxide content increases, leading to a decline in physiological function and resistance. This makes the fruit more vulnerable to pathogenic bacteria, ultimately impacting its post-harvest lifespan and preservation time. ‘Orin’ exhibited a CAT peak of 56.56 U g^−1^ at 56 days after storage, while ‘RX’ showed a CAT peak of 16.17 U g^−1^ at 84 days after harvest (Figure 4A).

Ascorbic acid (AsA) is a crucial antioxidant that helps in scavenging reactive oxygen species (ROS) and plays a significant role in maintaining the quality of fruits and vegetables. Upon harvest, ‘RX’ apples exhibit a notably higher ascorbic acid content compared to ‘Orin’ apples, measuring at 14.33 mg g^−1^. As storage time is prolonged, the ascorbic acid content in both groups gradually decreases. It is noteworthy that the ascorbic acid content of ‘RX’ consistently surpasses that of ‘Orin’, and that the ascorbic acid content of ‘RXH’ always exceeds that of ‘Envy’ (Figure 4B).

Peroxidase, an essential oxidoreductase found in plants, undergoes continuous changes in activity during growth, development, maturation, and senescence. Throughout the storage period, each variety exhibited a pattern of initially increasing and then decreasing peroxidase activity. ‘Orin’ displayed significantly higher overall peroxidase content compared to ‘RX’. With the exception of 42 d and 56 d, ‘Envy’ demonstrated notably higher peroxidase activity than ‘RXH’ (Figure 4C). 

Malondialdehyde (MDA) is a byproduct of membrane lipid peroxidation, and its levels directly indicate the extent of cytoplasmic membrane peroxidation. As depicted in Figure 4D, the MDA content trend across different cultivars during the entire storage period was largely consistent. ‘Orin’ exhibited the greatest increase in MDA content, reaching 2.1 times its initial storage level during the late storage period (Figure 4D). In contrast, ‘RX’ had significantly lower MDA content. Additionally, ‘RXH’’s MDA increase was notably lower than that of ‘Envy’.

### 3.3. Measurement of Cell-Wall-Degrading Enzyme Activities during Storage of Fruits of Four Cultivars

Throughout fruit storage, the trends in firmness changes were consistent across different cultivars, yet variations in firmness were observed at different storage periods. To investigate this further, we analyzed the enzyme activities associated with cell wall degradation in four cultivars. Our results showed that the activities of PG and CEL initially increased and then decreased during fruit storage, with the peak activity occurring at 42 days of storage (Figure 5A,B). Specifically, the PG and CEL activities of ‘RX’ fruit were notably lower than those of ‘Orin’ throughout the storage period. Moreover, the PG and CEL activities of ‘RXH’ remained consistently lower compared to ‘Envy’. Additionally, the PE activity of ‘Orin’ was significantly higher than that of other cultivars at harvest (Figure 5C), and this high level was maintained throughout the storage period. There were no significant differences in the PE activities of ‘Envy’ and ‘RXH’ during the later stages of storage. The β-Gal activity of all four cultivars exhibited an increasing trend during storage, with ‘Orin’ showing the highest activity at 56 days (32.42). Notably, the β-Gal activity of ‘RX’ was significantly lower than that of ‘Orin’ throughout the storage period, and the activity of ‘RXH’ was notably lower than that of ‘Envy’ (Figure 5D).

### 3.4. Variations in Cell Wall Composition of Four Cultivars of Fruit during Storage

The decline in fruit firmness is intricately linked to alterations in cell wall composition and cell expansion. To further investigate the changes in the four cultivars of fruits during storage, the levels of water-soluble pectin (WSP), ASP, and cellulose were assessed (Figure 6). Throughout the storage duration, the WSP content exhibited a general upward trajectory for each variety. ‘Orin’ displayed notably higher WSP levels compared to ‘RX’, while ‘Envy’ WSP content surpassed that of ‘RXH’ (Figure 6A). As the storage period progressed, the ASP content for each variety declined. ‘Orin’ experienced the most pronounced decrease in ASP content, with levels in the late storage period plummeting to 38.26% of the initial value. ‘Envy’ showed a more pronounced decline in ASP compared to ‘RXH’, which only decreased by 33.59% (Figure 6B). The fluctuation in cellulose content mirrored that of ASP, with ‘Orin’ exhibiting the most substantial decrease in cellulose content (Figure 6C).

### 3.5. Comparative Analysis of Cell Microstructure in Four Varieties of Fruits Was Conducted during Storage

Staining and observation of sections from fruit tissues revealed that, upon harvest, the pulp cells in each variety were closely arranged with minimal intercellular gaps (Figure 7). Notably, ‘Orin’ exhibited fewer cells compared to ‘RX’ under the same field of view. As the fruits softened and starch hydrolyzed, cell tension decreased, intercellular connections loosened, and gaps between cells increased, particularly evident in ‘Orin’ fruits. Throughout storage, ‘RX’ displayed relatively minor changes in intercellular space. In the later stages, ‘Envy’ and ‘RXH’ pulp cells also became irregular, with a significant portion nearly disintegrating. Variations in cell structure (size and arrangement) may contribute to the diverse textures observed in different fruit types.

### 3.6. Expression of Genes Related to Cell Wall Degradation during Fruit Storage in Four Cultivars

In our previous study, we analyzed the expression levels of cell wall hydrolase and ethylene-related genes in four different fruit varieties during the ripening process. Using qRT-PCR technology, we monitored the expression of nine genes over the storage period, as shown in Figure 8. Throughout storage, all nine genes exhibited a consistent increase in expression levels. Specifically, the expression of *MdPLs* in ‘RX’ remained lower compared to ‘Orin’. Moreover, the expression levels of *MdPLs* in ‘RXH’ were consistently lower than those in ‘Envy’. The expression patterns of *MdPG* and *MdXTH* were similar, with lower levels observed in ‘RX’ and ‘RXH’ compared to ‘Orin’ and ‘Envy’. Notably, the enzymes ACC synthases (ACSs) and ACC oxidases (ACOs) play crucial roles in ethylene synthesis [28], underscoring the importance of monitoring *MdACSs* and *MdACOs* expression. Over time, the expression of these genes significantly increased in all four varieties, albeit with a gentler upward trend in ‘RX’ compared to ‘Orin’. Among the varieties tested, ‘RXH’ exhibited the lowest expression level of *MdACS1*. The relative expression of *Mdβ-gals* in ‘Orin’ fruits increased gradually during storage and eventually stabilized in later stages. Similarly, the expression trend of *Mdβ-gals* in ‘RX’, ‘Envy’, and ‘RXH’ fruits mirrored that of ‘Orin’, but with higher levels observed in the former three varieties. Furthermore, the expression of MdEXP in ‘Orin’ significantly surpassed that of the other varieties. 

### 3.7. Correlation Analysis of Apple Fruit Traits, Cell Wall Components, Cell-Wall-Degrading Enzyme Activities, and Gene Expression

A notable inverse relationship was identified between apple fruit firmness and the expression of genes associated with cell wall degradation (Table 1). Furthermore, a strong positive correlation was found between *MdPL1* gene expression related to pectin and variables such as weight loss rate (r = 0.98), WSP content (r = 0.89), and PE content (r = 0.60). Conversely, a negative correlation was observed between *MdPL1* gene expression and cellulose content (r = −0.89), as well as between *MdPL2* gene expression and cellulose content (r = −0.92). The correlation coefficient between *MdPG* gene expression and PG enzyme activity was r = 0.70. Additionally, a significant negative correlation was noted between ASP content and the expression of each gene. Notably, *Mdβ-Gal2* gene expression displayed significant positive correlations with loss rate (r = 0.88), WSP (r = 0.96), and β-Gal (r = 0.65) activity, while exhibiting significant negative correlations with fruit hardness (−0.91) and ASP (−0.811) content.

## 4. Discussion

Fruit storage and preservation play a crucial role in fruit production, directly impacting the economic outcomes of the fruit industry. The rate of softening in apple fruit is a key factor influencing its marketability and shelf life. Various apple varieties demonstrate varying rates of softening during storage. This study utilized four apple varieties to investigate the fluctuation in cell wall component content and enzyme activities throughout fruit storage. Additionally, the research delved into the disparities in expression levels of pivotal genes involved in the cell wall metabolism process.

The storage trends of the four fruit varieties are generally similar, with some variations. The respiration rate and ethylene release are key factors influencing the fruit’s storage life. In the case of climacteric fruits like apples, the occurrence of respiratory climacteric indicates a boost in the fruit’s nutrient metabolism. Ethylene release, as a mature senescence hormone, serves as a direct indicator of the fruit’s metabolic condition [29]. Through our study comparing the respiration rates and ethylene release of the four fruit varieties during storage, we observed that the peak respiration rate occurred on the 28th day post-harvest, followed by a significant decrease. The ethylene release trend aligned with the respiratory rate changes (Figure 2). Subsequently, we monitored the soluble solids and titratable acid content of each variety throughout storage. Our results revealed an increasing trend in soluble solid content for all four varieties, stabilizing and showing significant differences in the later stages of storage (Figure 3).

Significant differences were observed in the fruit firmness of the four apple varieties on the day of harvest and throughout the storage period. ‘Orin’ fruits showed a faster decrease in firmness compared to ‘RX’, highlighting the poor storage stability of ‘Orin’ fruits (Figure 1A). ‘RXH’ exhibited significantly higher fruit firmness than the other tested varieties, indicating excellent storage stability. A similar trend was also noted in the detection of fruit weight loss rate (Figure 1B).

Starch, found in fruit cells, is crucial for maintaining cell turgor and fruit firmness. In apples, the starch content is a key indicator of harvest maturity [30]. During post-harvest storage, starch is converted into sugar by amylase, which is a vital respiratory substrate in fruits. This study observed a rise in respiration rate and ethylene release post-harvest, leading to a rapid increase in amylase activity. This, in turn, accelerated the hydrolysis of starch, raised soluble solids content, reduced cell turgor, and ultimately softened the fruit.

Numerous studies have demonstrated that enzymes involved in cell wall metabolism induce changes in the structure of the supporting cell wall, leading to increased cell permeability, leakage of cell fluid, and a decrease in fruit hardness, ultimately facilitating fruit ripening [27]. The softening process of fruit involves alterations in cell wall structure and degradation of various cell wall components [11]. Pectin, cellulose, and hemicellulose are arranged in a random manner, forming the polysaccharide network structure of the cell wall through hydrogen bonds, covalent bonds, and hydrophobic forces. In particular, pectin is organized within the cellulose and hemicellulose microfilaments of the cell wall, playing a crucial role in determining fruit texture. Fruit ripening is typically linked to the conversion of protopectin into water-soluble pectin, gradual loosening of cellulose and hemicellulose skeleton structures, and damage to the cell structure [31]. This study observed a decrease in fruit hardness across four varieties alongside an increase in WSP content and a decrease in ASP content (Figure 6A,B). This aligns with previous research on different fruits, suggesting that pectin degradation is a key factor in the softening of apple fruits. ‘Orin’ fruits exhibited a higher degree of pectin degradation compared to ‘RX’, leading to a relatively faster decrease in fruit firmness. Additionally, ‘RXH’ showed less pectin degradation compared to ‘Envy’ (Figure 6).

The degradation process of cell wall components is often facilitated by cell wall degradation-related enzymes, including polygalacturonase (PG), pectin methylesterase (PME), and glycosidases (β-Gal). This study observed that the activities of PE and PG in four fruit varieties were initially low during storage, but increased rapidly in the middle stage, leading to a significant reduction in fruit hardness. In the later stages of storage, PE activity remained high, while PG activity decreased, resulting in a gradual softening of the fruit. β-Gal plays a role in reducing galactose residues in the cell wall of mature fruits, leading to pectin solubilization and destabilization of the cell wall structure, ultimately causing fruit softening. The findings indicate that ‘Orin’ fruits exhibit low β-Gal activity in the early stages of storage, but experience a rapid increase in the middle and late stages, surpassing the other three apple varieties in terms of β-Gal activity (Figure 5D). Gwanpua discovered that the β-Gal activity in ‘Golden Delicious’ apples, which softened rapidly, was notably higher compared to ‘Kanzi’ apples, which softened slowly. However, there was no significant difference in PE activity between the two apple varieties [28]. The increase in CEL activity will result in the depolymerization of cellulose, leading to the disintegration of the cell wall by degrading cellulose and hemicellulose. Higher CEL activity correlates with a faster decrease in fruit hardness. Among the four varieties tested in this study, ‘Orin’ exhibited the highest CEL activity, while ‘RXH’ and ‘RXH’ showed consistently low levels of CEL activity (Figure 5B).

Utilizing qRT-PCR technology, researchers can elucidate the expression patterns of enzymes involved in cell wall degradation across various fruit varieties. This sets the stage for further exploration into the mechanisms influencing fruit firmness and a deeper understanding of the ripening process. Our study revealed that, as the storage period was prolonged, the relative expression levels of PL and PG significantly increased in all four varieties. However, the extent and timing of up-regulation varied, leading to distinct hardness characteristics during storage. Furthermore, the transcription levels of key genes associated with fruit ripening, such as *MdACO1* and *MdACS1*, exhibited gradual increments with differing rates of up-regulation among the varieties. Notably, we observed a significant negative correlation between apple fruit firmness and the expression of genes involved in ethylene synthesis and hydrolase within the cell wall metabolism pathway. A notable positive correlation was observed between WSP content and the expression levels of *MdPL1*, *MdPL2*, and other genes, suggesting the involvement of WSP in the fruit softening process during storage. Conversely, the cellulose content exhibited a significant negative correlation with genes associated with cell wall degradation.

## 5. Conclusions

The texture and storage properties of different apple varieties exhibit significant variation. The rate at which fruit hardness decreases post-harvest plays a crucial role in determining consumer satisfaction and shelf life. This decline is closely linked to the metabolism of cell wall polysaccharides. As such, this research study delved into the physiological, biochemical, and structural changes, as well as the dynamic molecular alterations, in four distinct apple varieties during storage. During storage, the activity of cell wall hydrolases and the relative expression levels of related genes increase by regulating the degradation of cell wall materials. This indicates that the decrease in apple fruit hardness during storage is caused by the activity of cell wall hydrolases and the relative expression levels of related genes. Apple fruits of different varieties exhibit differences in WSP, ASP and cellulose content, PG activity, and relative expression of *Mdβ-Gal*, which may explain the variations in their texture.

## Figures and Tables

**Figure 1 foods-13-01563-f001:**
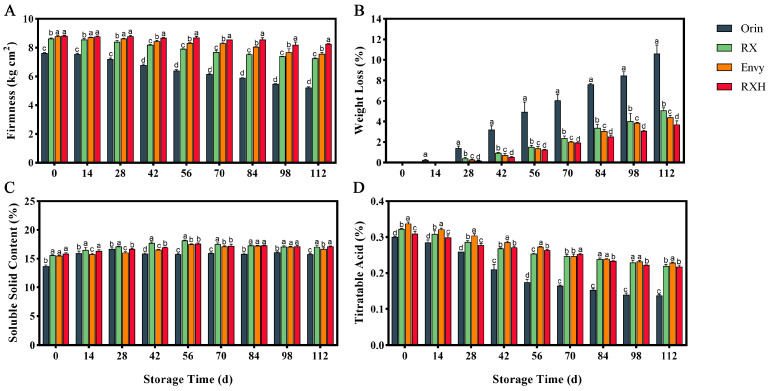
Changes in firmness (**A**), water loss rate (**B**), SSC (**C**), and TA (**D**) during fruit storage of four varieties. Error bars indicate SD (*n* = 3). Different lowercase letters in columns indicate significant differences between sampling cultivars for each date by Duncan’s test (*p* < 0.05).

**Figure 2 foods-13-01563-f002:**
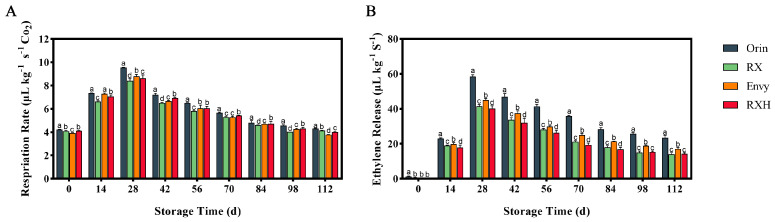
Changes in respiration rate (**A**) and ethylene release (**B**) of four varieties during storage. Error bars indicate SD (*n* = 3). Different lowercase letters in columns indicate significant differences between sampling cultivars for each date by Duncan’s test (*p* < 0.05).

**Figure 3 foods-13-01563-f003:**
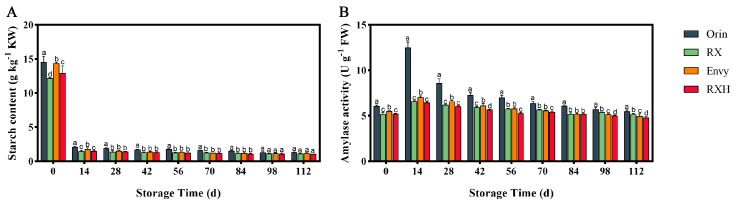
Changes of starch content (**A**) and amylase activity (**B**) in fruits of four varieties during storage. Error bars indicate SD (*n* = 3). Different lowercase letters in columns indicate significant differences between sampling cultivars for each date by Duncan’s test (*p* < 0.05).

**Figure 4 foods-13-01563-f004:**
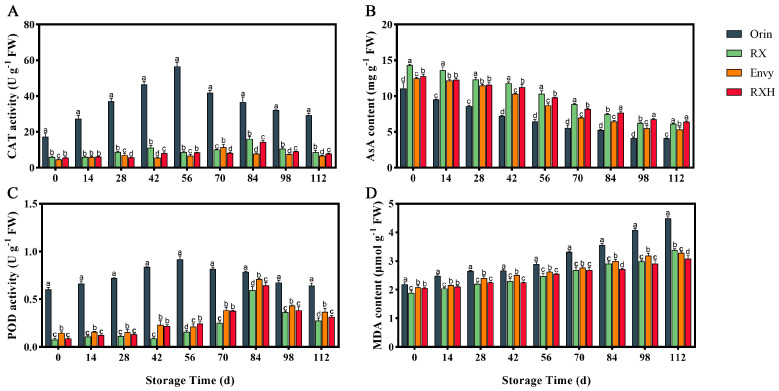
Changes in CAT (**A**), AsA (**B**), POD (**C**), and MDA (**D**) activities of four varieties during fruit storage. Error bars indicate SD (*n* = 3). Different lowercase letters in columns indicate significant differences between sampling cultivars for each date by Duncan’s test (*p* < 0.05).

**Figure 5 foods-13-01563-f005:**
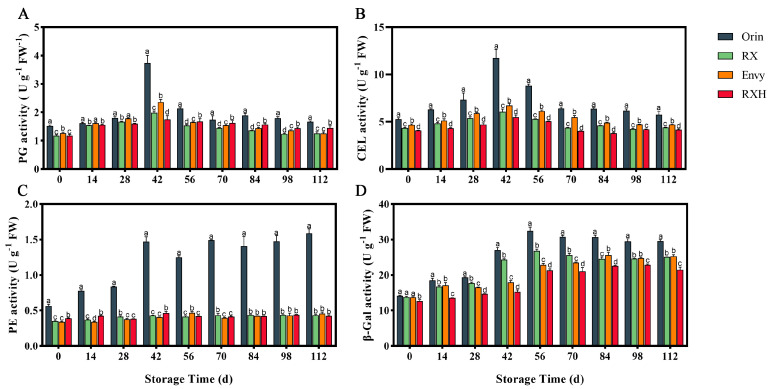
Changes in PG (**A**), CEL (**B**), PE (**C**), and β-Gal (**D**) activities of four varieties during fruit storage. Error bars indicate SD (*n* = 3). Different lowercase letters in columns indicate significant differences between sampling cultivars for each date by Duncan’s test (*p* < 0.05).

**Figure 6 foods-13-01563-f006:**
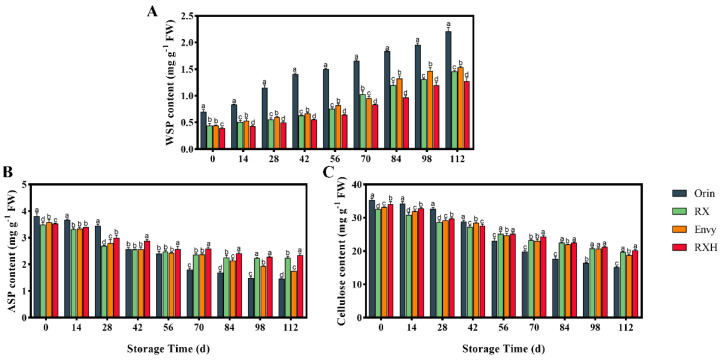
Changes of WAP (**A**), ASP (**B**), and cellulose (**C**) contents during storage of four varieties. Error bars indicate SD (*n* = 3). Different lowercase letters in columns indicate significant differences between sampling cultivars for each date by Duncan’s test (*p* < 0.05).

**Figure 7 foods-13-01563-f007:**
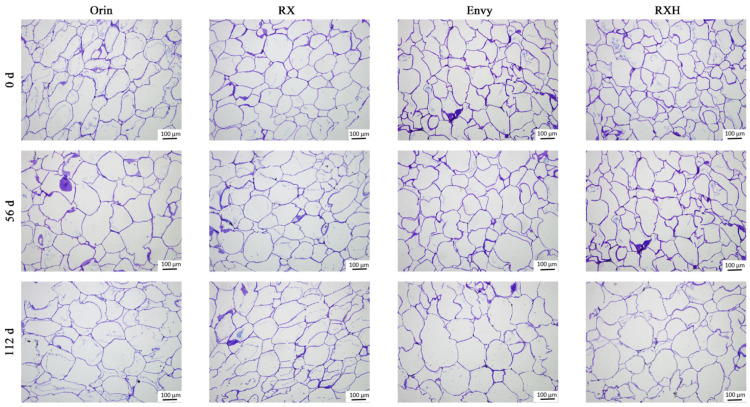
Comparison of the changes of fruit pulp cell microstructure during storage of four varieties. Images were obtained at 100× magnification (bar, 100 μm).

**Figure 8 foods-13-01563-f008:**
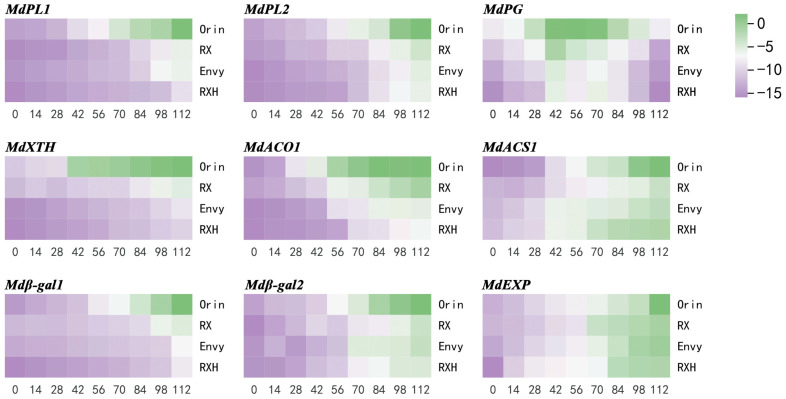
Expression patterns of cell wall degradation enzymes genes of four cultivars.

**Table 1 foods-13-01563-t001:** Correlation analysis of cell wall components and hydrolase activities with gene expression.

Index	*MdPL1*	*MdPL2*	*MdPG*	*MdXTH*	*MdACO1*	*MdACS1*	*Mdβ-Gal1*	*Mdβ-Gal2*	*MdEXP*
Firmness	−0.91	−0.91	0.04	−0.80	−0.99	−0.97	−0.76	−0.91	−0.97
Loss rate	0.988	0.98	−0.13	0.93	0.94	0.93	0.89	0.88	0.93
WSP	0.89	0.91	−0.01	0.80	0.97	0.97	0.74	0.96	0.91
ASP	−0.61	−0.66	−0.034	−0.50	−0.79	−0.77	−0.46	−0.81	−0.71
Cellulose	−0.89	−0.92	0.02	−0.79	−0.99	−0.96	−0.76	−0.93	−0.92
PG	−0.15	−0.15	0.70	−0.15	−0.10	−0.07	−0.17	−0.02	−0.13
CEL	−0.1672	−0.1573	0.569	−0.1648	−0.117	−0.0772	−0.1972	−0.04	−0.20
PE	0.6026	0.6515	0.0355	0.4955	0.743	0.7742	0.4424	0.78	0.62
β-Gal	0.475	0.4972	0.1081	0.3813	0.7009	0.7076	0.3259	0.65	0.56

## Data Availability

The original contributions presented in the study are included in the article/Appendix A, further inquiries can be directed to the corresponding author.

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
