# Peer review of "Comparison of Fruit Texture and Storage Quality of Four Apple Varieties"

_foods, 2024, doi:10.3390/foods13101563_

Round 1

Reviewer 1 Report

Comments and Suggestions for Authors

The authors evaluated an interesting topic, about the fruit texture and storage quality of apples with this article entitled Comparison of fruit texture and storage quality of four apple varieties.

The paper is well written and well organized.

Some minor remarks are following.

In the abstract please give the whole names of the acronyms WSP and CEL.

Line 29-31.In recent years, the gradual quality deterioration of apple fruits during storage has emerged as a key concern [5], Fruit texture, being a crucial element of fruit quality, directly impacts the fruit's taste and shelf life. Please rewrite this sentence.

Line 104-105.Please provide the geographic coordinates.

Line 121.mm s-1. Please write again in the right way.

All figures and tables should appear in the text immediately after their reference.

Author Response

Dear Editors and Reviewers:

Thank you for your letter and for the reviewers’ comments concerning our manuscript entitled “Comparison of fruit texture and storage quality of four apple varieties” (ID: foods-2993183). Those comments are all valuable and very helpful for revising and improving our paper, as well as the important guiding significance to our researches. We have studied comments carefully and have made correction which we hope meet with approval. Revised portion are marked in red in the paper. The main corrections in the paper and the responds to the reviewer’s comments are as flowing: Responds to the reviewer’s comments:

Reviewer #1:

1): Thank you very much. We changed it in Line 16.

2): Thanks for your suggestions. We changed it in Line 29-30. “Fruit texture, as an important part of fruit quality, directly determines the taste and storage of fruit.”

3): Thank you very much. We changed it in Line 106-108. “The experiment was conducted in the Bai Shui Apple Experimental Station (35°02′N, 109°06′E; 908 m altitude) of Northwest A&F University.”

4): Thank you very much. We are very sorry for our incorrect writing. We changed it in Line 125.

5): Thanks for your suggestions. We changed it in Line 210-214, 223-226, 235-238, 265-268,285-288,302-305, 317-319, 338-340,355-357.

Special thanks to you for your good comments.

Reviewer 2 Report

Comments and Suggestions for Authors

The manuscript presents an extensive set of research results on the properties of apples during storage. The Authors compared four apple varieties, assessing, among others, their physiological characteristics (firmness, weight loss, starch content, ethylene release, etc.), antioxidant activity, condition of cell wall polysaccharides, activity of enzymes related to cell wall degradation, cell wall morphology and the expression of selected genes, additionally. The scope of research was very extensive, but the manuscript lacks an explanation of the research assumptions and a clear indication of the purpose of the comparison.

First, the Authors should present the purpose of their research and the criteria for selecting the evaluated apple varieties. The manuscript contains a general description of the "Orin" and "RX" varieties, but does not describe the "Envy" and "RXH" at all.

The abstract should not contain unexplained abbreviations. Some of them are not explained throughout the article.

Then in the Introduction, especially the second paragraph (lines 47 to 68), there is a lot of repetitive content. Authors should indicate what species of fruit they are writing about when they indicate the mechanisms of ripening and degradation that take place in them. Here, general relationships for all fruits are alternately quoted, interspersed with research results on specific species or varieties. This is difficult for the reader to follow and requires reorganization of the text.

There are also minor shortcomings, e.g. in line 78 the researcher's name ("Sebastian") is given instead of his surname ("Wolf"), or lack of explanation of the abbreviations used, e.g. in line 143.

A very extensive presentation of the research results is followed by a very short discussion of them. The Authors should emphasize what correlations were observed by the assessment of selected properties of apples and their changes during storage. There is only a short section in the manuscript devoted to the analysis of the data presented in Table 1.

The conclusions drawn from the study should be clearly presented in the last section of the manuscript. Now section 5. “Conclusions” mainly consists of very general statements. The Authors should explain more clearly the conclusions of their research and their comparison with the current state of knowledge.

Author Response

Dear Editors and Reviewers:

Thank you for your letter and for the reviewers’ comments concerning our manuscript entitled “Comparison of fruit texture and storage quality of four apple varieties” (ID: foods-2993183). Those comments are all valuable and very helpful for revising and improving our paper, as well as the important guiding significance to our researches. We have studied comments carefully and have made correction which we hope meet with approval. Revised portion are marked in red in the paper. The main corrections in the paper and the responds to the reviewer’s comments are as flowing: Responds to the reviewer’s comments:

Reviewer #2:

1):We have made correction according to the Reviewer’s comments. We have re-written this part according to the Reviewer’s suggestion. We changed it in Line 79-88.

2):Thanks for your suggestions. We changed it in Line 16.

3):Thank you very much. We changed it in Line 49-58.

4):Thank you very much. We changed it in Line 68.

5):Thanks for your suggestions. We addeded it in Line 437-441. “A notable positive correlation was observed between WSP content and the expression levels of MdPL1, MdPL2, and other genes, suggesting the involvement of WSP in the fruit softening process during storage. Conversely, the cellulose content exhibited a significant negative correlation with genes associated with cell wall degradation.”

6): Thank you very much. We changed it in Line 448-454. “During storage, the activity of cell wall hydrolases and the relative expression levels of related genes increase by regulating the degradation of cell wall materials. This indicates that the decrease in apple fruit hardness during storage is caused by the activity of cell wall hydrolases and the relative expression levels of related genes. of. Fruits of different varieties have differences in WSP, ASP and cellulose content, PG activity and relative expression of Mdβ-Gal. These differences may be the reason for the differences in texture of apple fruits of different varieties.”

Special thanks to you for your good comments.

Reviewer 3 Report

Comments and Suggestions for Authors

The subject is interesting and current and can contribute to the literature. According to the journal rules, the manuscript has all the necessary sections. On the other hand, some points should be improved, listed below;

The discussion section has one reference, and it belongs to writers. The section should contain references. The discussion of the results should be compared with the literature.

-The conclusion section should be enlarged

-The graphical abstract can be formed.

-The manuscript has one table related to results. Other results should be given in the manuscript or as supplementary.

Author Response

Dear Editors and Reviewers:

Thank you for your letter and for the reviewers’ comments concerning our manuscript entitled “Comparison of fruit texture and storage quality of four apple varieties” (ID: foods-2993183). Those comments are all valuable and very helpful for revising and improving our paper, as well as the important guiding significance to our researches. We have studied comments carefully and have made correction which we hope meet with approval. Revised portion are marked in red in the paper. The main corrections in the paper and the responds to the reviewer’s comments are as flowing: Responds to the reviewer’s comments:

Reviewer #3:

1): Thank you very much. We changed it in Line 371, 386, 394, 401, 421, and 539-549.

2): Thanks for your suggestions. We changed it in Line 448-454.

3): Thank you very much. We added a graphical abstract.

4): Thank you very much. We changed it in Line 210-214, 223-226, 235-238, 265-268,285-288,302-305, 317-319, 338-340,355-357.

Special thanks to you for your good comments.

Round 2

Reviewer 2 Report

Comments and Suggestions for Authors

I appreciate the Authors' responses to the comments in the review and the changes introduced in the article. The purpose of the research as well as the results and conclusions obtained are presented now in a much more explicit manner. The errors indicated in the previous revision have also been corrected. In my opinion, the article in its current form may be considered for publication.